# Overview of Aerosol Properties in the European Arctic in Spring 2019 Based on In Situ Measurements and Lidar Data

Fieke Rader [1], Rita Traversi [2,3], Mirko Severi [2,3], Silvia Becagli [2,3], Kim-Janka Müller [1], Konstantina Nakoudi [1,4] and Christoph Ritter [1,*]

[1] Helmholtz Centre for Polar and Marine Research, Alfred Wegener Institute, Telegrafenberg A45, 14473 Potsdam, Germany; fieke.rader@awi.de (F.R.); kim.janka.mueller@alumn.uni-oldenburg.de (K.-J.M.); konstantina.nakoudi@awi.de (K.N.)

[2] Department of Chemistry, University of Florence, Via della Lastruccia 3, 50019 Sesto Fiorentino, Florence, Italy; rita.traversi@unifi.it (R.T.); mirko.severi@unifi.it (M.S.); silvia.becagli@unifi.it (S.B.)

[3] Institute of Polar Science, ISP-CNR, Via Torino, 155, 30172 Venezia Mestre, Venice, Italy

[4] Institute of Physics and Astronomy, University of Potsdam, Karl-Liebknecht 24/25, 14476 Potsdam, Germany

[*] Correspondence: christoph.ritter@awi.de

**Abstract:** In this work, we analysed aerosol measurements from lidar and $PM_{10}$ samples around the European Arctic site of Ny-Ålesund during late winter–early spring 2019. Lidar observations above 700 m revealed time-independent values for the aerosol backscatter coefficient ($\beta$), colour ratio (CR), linear particle depolarisation ratio ($\delta$) and lidar ratio (LR) from January to April. In contrast to previous years, in 2019 the early springtime backscatter increase in the troposphere, linked to *Arctic haze*, was not observed. In situ nss-sulphate (nss-$SO_4^{2-}$) concentration was measured both at a coastal (Gruvebadet) and a mountain (Zeppelin) station, a few kilometres apart. As we employed different measurement techniques at sites embedded in complex orography, we investigated their agreement. From the lidar perspective, the aerosol load (indicated by $\beta$) above 700 m changed by less than a factor of 3.5. On the contrary, the daily nss-$SO_4^{2-}$ concentration erratically changed by a factor of 25 (from 0.1 to 2.5 ng m$^{-3}$) both at Gruvebadet and Zeppelin station, with the latter mostly lying above the boundary layer. Moreover, daily nss-$SO_4^{2-}$ concentration was remarkably variable (correlation about 0.7 between the sites), despite its long-range origin. However, on a seasonal average basis the in situ sites agreed very well. Therefore, it can be argued that nss-$SO_4^{2-}$ advection mainly takes place in the lowest free troposphere, while under complex orography it is mixed downwards by local boundary layer processes. Our study suggests that at Arctic sites with complex orography ground-based aerosol properties show higher temporal variability compared to the free troposphere. This implies that the comparison between remote sensing and in situ observations might be more reasonable on longer time scales, i.e., monthly and seasonal basis even for nearby sites.

**Keywords:** Arctic haze; aerosol measurements; aerosol properties; in situ aerosol measurements; aerosol remote sensing; lidar; Svalbard

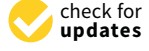



## 1. Introduction

The Arctic is known to be a meteorologically sensitive region as its near-surface temperature increases at least twice as fast as in the rest of the globe. This phenomenon is called *Arctic Amplification* [1,2] and may have implications also for the mid-latitudes [3]. Especially, the Svalbard-Barents Sea region currently faces a pronounced winter warming of almost three degrees per decade, part of which seems to be caused by more efficient advection of north Atlantic air masses [4]. However, the feedback mechanisms of *Arctic Amplification* are not yet fully understood, with climate models disagreeing on the sign of the total Arctic radiative feedback [5] and the contribution of aerosol (by direct radiative effect or as precursors of clouds) remaining uncertain [6]. While the assessment of Arctic aerosol

radiative properties in terms of case studies [7], regional [8] and global [9] models makes progress, a systematic comparison between modelled and quality-assured observational aerosol data is still missing. At the same time, meteorological data from the Arctic are sparse. The inclusion of additional data sets (e.g., radiosounding from dedicated campaigns) has led to improved weather forecasts in boreal regions [10], but air mass backtrajectory calculations and, hence, the aerosol origin, remain insecure in the Arctic [11].

The most distinct pattern of accumulation mode aerosol in the Arctic occurs during spring in the free troposphere and forms the so-called *Arctic haze* [12–14]. Antropogenic sulphate is a key *Arctic haze* component [15], while at coastal sites sea salt and biogenic contributions are also significant for the sulphate budget [16,17]. Since the early 1980s sulfate concentration showed a decreasing trend over the Arctic ($-2$–3% year$^{-1}$, 1980–2010) [18], including the Zeppelin station ($-1.3 \pm 1.2$ ng m$^{-3}$ year$^{-1}$, 1990–2008) [19]. In terms of aerosol scattering coefficient, an increasing trend (+0.05 Mm$^{-1}$ year$^{-1}$) is only observed during summer over Zeppelin station, with statistically insignificant trends in the other seasons [20]. At the same time, the intensity of *Arctic haze* has decreased over Ny-Ålesund in the last two decades [21] and nowadays it is comparable to biomass-burning events entering the Arctic during summer [22–24]. However, the individual contribution of aerosol sources, sinks and transport pathways to the changing Arctic aerosol patterns has not been fully constrained.

Ny-Ålesund is an international research village on the west coast of Spitsbergen Island, Svalbard archipelago, (78.9° N, 11.9° E) in the European part of the Arctic. Aerosol properties are monitored in situ in the Gruvebadet station [16] (about 1 km southwest of the village) and on the mountain Zeppelin station (2.5 km to the south at 474 m above sea level (m a.s.l.)) [25] as well as by means of remote sensing on a long-term basis [23,26,27]. In recent years, noteworthy efforts focused on combining aerosol measurements from different techniques around Ny-Ålesund. Tesche et al. [28] systematically assessed the agreement of aerosol extinction between CALIOP spaceborne lidar and in situ observations from Zeppelin station. The highest agreement was achieved in terms of probing similar air masses rather than relying on the closest satellite overpass distance. Ferrero et al. [29] systematically analysed vertical profiles of aerosol in the lowest 800 m over Ny-Ålesund by means of tethered balloon in situ measurements. The authors reported different springtime regimes of aerosol vertical distribution in connection with different coupling between surface and boundary layer aerosol conditions. In a follow-up study (Ferrero et al. [30]), aerosol measurements over Gruvebadet (in situ obtained from tethered balloon) and Ny-Ålesund (lidar-based) were combined. A successful closure was reached in terms of aerosol backscatter coefficient after complete aerosol chemical speciation.

Despite all efforts so far, there are open research questions: Given the mentioned climate change in Spitsbergen, what are long-term changes in properties of Arctic aerosol, both from remote sensing and ground-based in situ perspective? Second, as an aerosol closure in terms of chemical and optical properties typically does not provide a clear match for case studies, how do aerosol properties from two nearby in situ stations and a lidar compare on a seasonal scale?

In this study, we assess measurements from three nearby stations (AWIPEV, Gruvebadet and Zeppelin) spanning from ground-level to the free troposphere. Our first goal is to present lidar data and non-sea-salt sulphate (nss-SO$_4^{2-}$) concentration for the complete spring 2019 season and to shed some light on the aerosol properties. As a second goal, we contribute to the open question whether quasi-contemporaneous aerosol measurements from different measurement sources are reconcilable.

Therefore, the complex orography of Ny-Ålesund that introduces various micrometeorological phenomena [31] has to be considered. Under these conditions, the synthesis of aerosol products from different measurement techniques may be challenging even for neighbouring locations. Further, lidar only provides reliable information above the so-called complete *overlap height* [32], which is 700 m for our study. Moreover, a hygroscopic effect bias can be introduced between lidar and in situ [33]. Lidar probes aerosol at ambient

humidity conditions, whereas aerosol is sampled in situ under dry conditions. In this study, we thoroughly investigate the effect of hygroscopicity on the derived aerosol properties.

## 2. Instruments and Evaluation Methods

In this section, we briefly describe the instruments and main evaluation steps. Data from a Raman lidar in the research village as well as in situ aerosol data from Gruvebadet station (~700 m S-SW of the village, at almost sea level) and from the Zeppelin station (~2.5 km to the south at 474 m a.s.l.) are used. These three stations are marked by blue dots and the numbers 1 to 3, respectively, in Figure 1.

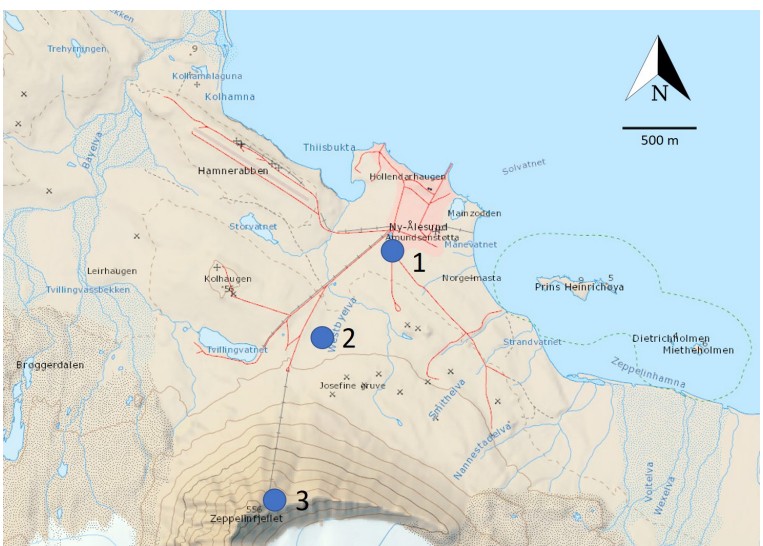

**Figure 1.** Site map of Ny-Ålesund. The locations of the Raman Lidar (1), the Gruvebadet Station (2) and the Zeppelin Station (3) are marked in the figure.

### 2.1. Lidar Data and Evaluation

The remote sensing data in this work were obtained by the Koldewey Aerosol Raman Lidar (KARL) system, which is installed in the Alfred Wegener Institute—Institute Paul Emile Victor (AWIPEV) research base. KARL consists of a 50 Hz Nd:YAG laser emitting at the three colours of 355 nm, 532 nm and 1064 nm at least 10 W each. A 70 cm receiving telescope, operating at 2.28 mrad field of view, collects apart from the aforementioned wavelengths also the inelastically scattered light at 387 nm, 607 nm (for the retrieval of the aerosol extinction) and 407 nm (for the absolute humidity). In our system, complete overlap (laser beam fully in the field of view of the receiving telescope) is achieved at 700 m above ground level (which in this case is almost the same as a.s.l.) and with a negligible difference as this is smaller than the height resolution of the lidar measurements (30 m). More details on the lidar system are given in Hoffmann [34]. In this work, 1295 lidar profiles were considered in total between the 9 January and the 30 April 2019 with 10 min and 30 m resolution.

In Ny-Ålesund, at least one Vaisala RS-41 radiosonde was launched daily during that period. During the Year of Polar Prediction (YOPP) campaign in February and March 2019, four radiosoundings were available daily. Radiosonde air density profiles were used for calculating the molecular scattering. Furthermore, radiosonde relative humidity (RH) profiles were used within 30 min windows around available lidar observations in order to investigate possible aerosol hygroscopic effects as in Müller et al. [35].

The lidar data have been analysed according to Ansmann [36]. The following quantities will be shown: the (volumetric) aerosol backscatter coefficient $\beta^{aer}$, units ($\mathrm{m}^{-1}\,\mathrm{sr}^{-1}$),

the linear particle depolarisation ratio ($\delta$), the lidar ratio and the colour ratio. $\delta$ is given by Equation (1):

$$\delta = \frac{\beta_\perp^{aer}}{\beta_\parallel^{aer}} \tag{1}$$

where $\beta_\perp^{aer}$ denotes the perpendicular-polarised aerosol backscatter coefficient and $\beta_\parallel^{aer}$ denotes the parallel-polarised aerosol backscatter coefficient. Spherical particles do not change the state of polarization in backscatter [37]. Therefore, $\delta$ is a measure of the asphericity of the aerosol. For the used lidar, the Rayleigh scattering of clear air has a depolarisation of 1.4% [37].

The lidar ratio (LR) is given by the ratio of aerosol extinction ($\alpha$) over aerosol backscatter coefficients (Equation (2)):

$$LR = \frac{\alpha^{aer}}{\beta^{aer}} \tag{2}$$

Finally, the colour ratio (CR) is defined as the ratio of aerosol backscatter coefficient in two different wavelengths (Equation (3)):

$$CR(\lambda_1, \lambda_2) = \frac{\beta_{\lambda_1}^{aer}}{\beta_{\lambda_2}^{aer}} \tag{3}$$

A strong wavelength dependency of the backscatter (large colour ratio) indicates small particles while for large particles the colour ratio converges towards unity (grey approximation).

KARL does not run on a 24/7 because thick low-level clouds give a too large backscatter that may damage the detectors. Therefore, it is mainly an aerosol and optically-thin cloud instrument. In our analysis, we removed clouds in order to have statistics on pure aerosol events. For consistency we used the same cloud threshold as in the iAREA study [27]. Data points with aerosol backscatter three times larger than the Rayleigh backscatter and CR (355 nm, 532 nm) smaller than 1.5 were removed.

### 2.2. In-Situ Measurements

In this work, we present sulphate concentration data in $PM_{10}$ aerosol from Gruvebadet (78.918° N, 11.895° E) and Mt. Zeppelin (78.908° N, 11.881° E) observatories (Figure 1). The Zeppelin Observatory is owned and managed by the Norwegian Polar Institute and is part of the Global Atmospheric Watch network. The monitoring of aerosol chemical composition at Gruvebadet began in 2010 and it is currently ongoing; aerosol samples were collected in the spring–summer period at different resolution (1 to 2 days) until 2018; since winter 2018/2019, all year-round samplings were started and they are continuing to date in a resolution of 2 days. The data set here presented refers to $PM_{10}$ aerosol samples collected on 47 mm diameter PTFE filters using a low volume inertial sampler (TECORA Skypost, Italy). The filters were prepared under a laminar flow hood in Florence and shipped to Ny-Ålesund; after sampling, the filters were stored in a freezer at "Dirigibile Italia" station and then shipped back to Florence together with field blanks. The filters were cut into two parts: one half was analysed for metals (as reported in Giardi [38]) or archived. The $PM_{10}$ mass was determined by weighing the filter before and after the sampling by means of a 5-digit microbalance (Sartorius ME235P). The filters were conditioned for 48 h (25 °C and 50% RH) before weighing. The portion of the filter devoted to chemical analysis was extracted in about 10 mL of ultrapure water (18 M$\Omega$ cm, Millipore MilliQ grade) in ultrasonic bath for 20 min. Sulphate and other major (Sodium, Calcium) and minor (MSA) ions which were used to calculate non sea salt sulphate (nss-$SO_4^{2-}$) and to assess sulphate source apportionment were determined by Ion Chromatography. The detailed procedure is described elsewhere [39].

For all the measured parameters reproducibility was better than 5% and filter blanks were lower than the detection limit. Detection limits for sulphate are 0.08 ng m$^{-3}$ taking

into account the most conservative conditions of sampled volume, i.e., 55 m$^3$ for daily resolution [38].

The sulphate source apportionment was accomplished by using a method successfully applied to samples collected in Ny-Ålesund in 2014 [40]. It is based on the use of a specific univocal marker of each source contributing to sulphate total budget and the corresponding marker-to-sulphate ratio characteristic of the source. Sulphate in the aerosol has five main sources: sea salt, crustal, marine biogenic, volcanic and anthropogenic. We have used Na$^+$, Ca$^{2+}$ and methanesulphonate (MSA) for sea salt, crustal and biogenic sources, respectively. As Na$^+$ and Ca$^{2+}$ have both seawater and crustal sources, in order to quantify the sea salt (ss-) and non-sea salt (nss-) fractions of Na$^+$ and Ca$^{2+}$ in every sample, we used a four-equation system [41,42]. The sea salt (ss-SO$_4^{2-}$) and crustal fractions of sulphate (cr-SO$_4^{2-}$) were then calculated by multiplying the ss-Na$^+$ and nss-Ca$^{2+}$ concentrations by 0.253 (SO$_4^{2-}$/Na$^+$ $w/w$ ratio in seawater) and 0.592 (SO$_4^{2-}$/Ca$^{2+}$ $w/w$ ratio in the uppermost Earth crust), respectively [43]. The remaining non-sea salt non-crustal fraction (nss-nc-SO$_4^{2-}$) fraction was calculated by subtracting from the total sulphate concentration (tot-SO$_4^{2-}$) the sea salt and the crustal contribution. The anthropogenic and the biogenic fractions were then included in the remaining nss-nc-SO$_4^{2-}$-fraction. While source markers of anthropogenic emissions are difficult to interpret, the contribution of the biogenic fraction (bio-SO$_4^{2-}$), which arises from from marine phytoplanctonic activity (via atmospheric oxidation of dimethylsulfide) was estimated by multiplying MSA, as univocal marker of marine biogenic emissions, concentration by a factor extrapolated from the relationship between nss-nc-SO$_4^{2-}$/MSA ($w/w$) ratio and MSA.

Regarding the measurements at Zeppelin site, sampling and analytical determination were accomplished by using the methods described in the EMEP Manual v1996 as reported in the EBAS NILU website and data were obtained from the same website.

## 3. Results

### 3.1. Lidar-Derived Aerosol Optical Properties in January–April 2019

In this subsection, we investigate the temporal variability of aerosol optical properties in the troposphere from January to April as well as their vertical variability. In Figures 2 and 3, aerosol backscatter coefficient and aerosol depolarisation histograms are given for the height interval 700 m to 1500 m. Backscatter values above $0.6 \times 10^{-6}$ m$^{-1}$ sr$^{-1}$ were hardly observed and the occurrence of increased aerosol backscatter was lower during the *Arctic haze* season (March, April) compared to late winter (January, February). Moreover, we did not observe any increase in the aerosol depolarisation during spring 2019. This unexpected, missing temporal variability of aerosol properties was also true for other height intervals (not shown for brevity). This indicates that the lidar-derived aerosol properties from January to April 2019 were mainly a function of altitude without pronounced temporal variability. The vertical distribution of aerosol optical properties was also analysed, with the relative frequencies of aerosol backscatter at 532 nm (Figure 4), colour ratio (Figure 5), depolarisation at 532 nm (Figure 6) and lidar ratio at 355 nm (Figure 7) displayed at four different altitude intervals.

During the whole period of late winter–early spring 2019, the highest backscatter (between $0.2 \times 10^{-6}$ and $0.5 \times 10^{-6}$ m$^{-1}$ sr$^{-1}$) was observed between 700 and 1500 m, exhibiting a large spread. This indicates higher aerosol content and variability in the lower altitudes compared to higher altitudes. Higher up in the atmosphere both the backscatter coefficient and its variability decreased.

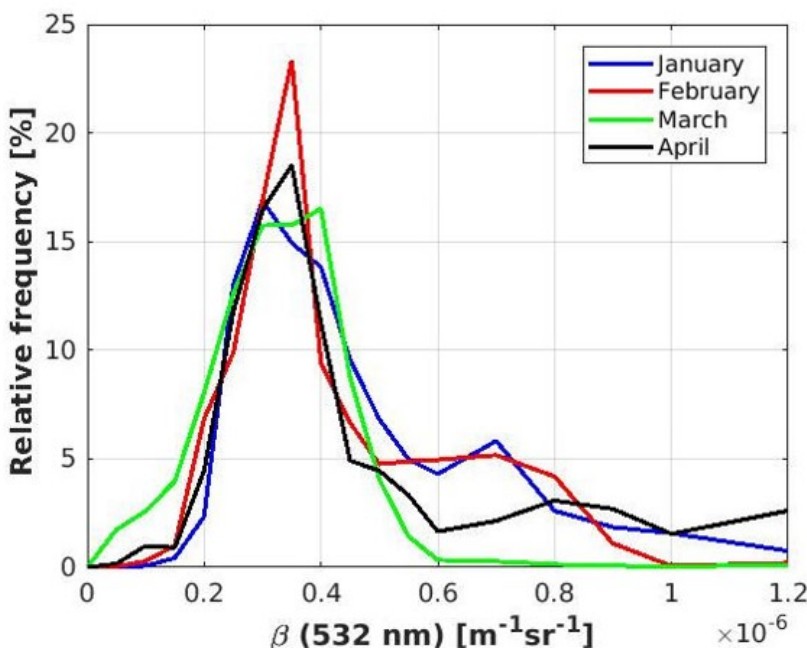

**Figure 2.** 700–1500 m: Frequency of $\beta_{532}$.

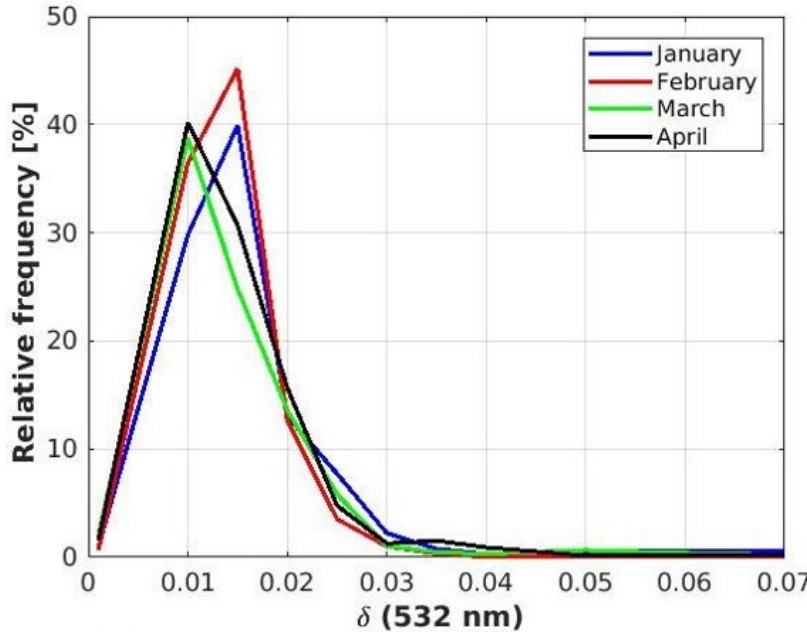

**Figure 3.** 700–1500 m: Frequency of $\delta_{532}$.

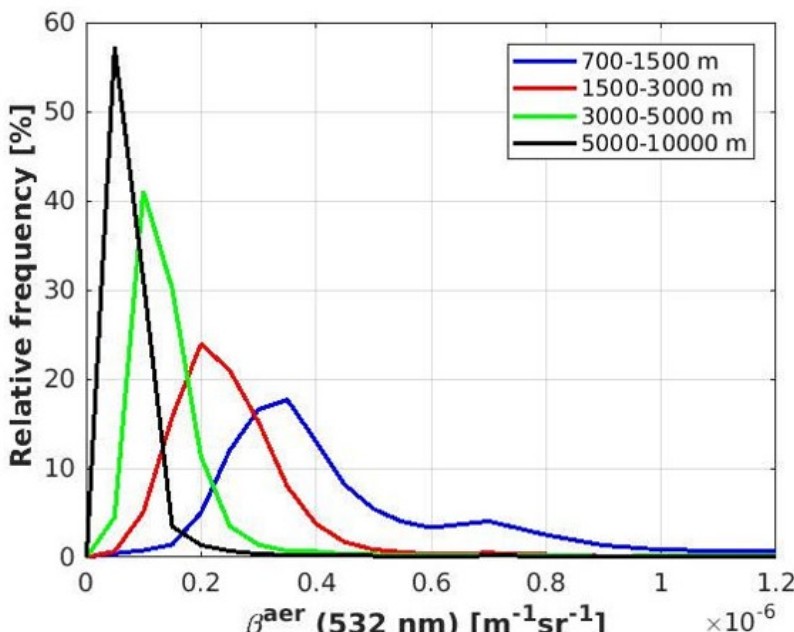

**Figure 4.** January–April: Frequency of $\beta_{532}$.

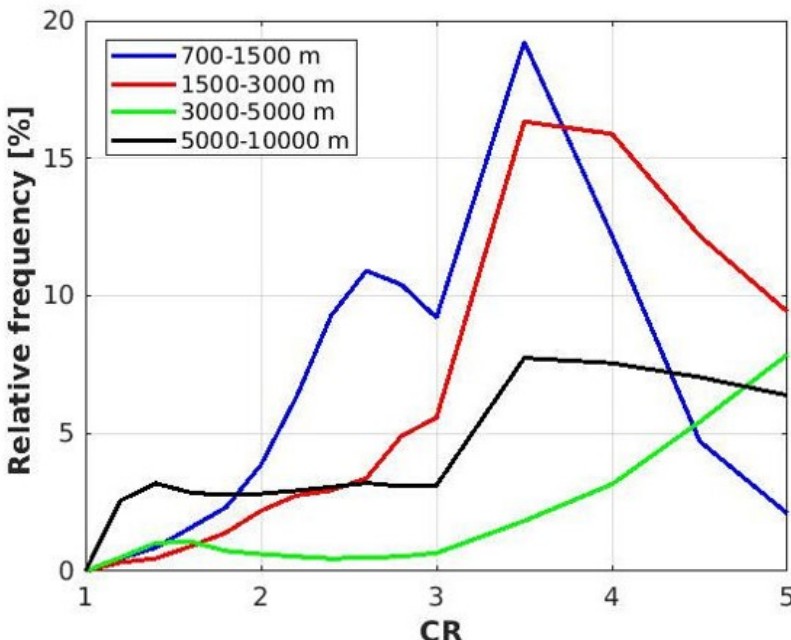

**Figure 5.** January–April: Frequency of the CR.

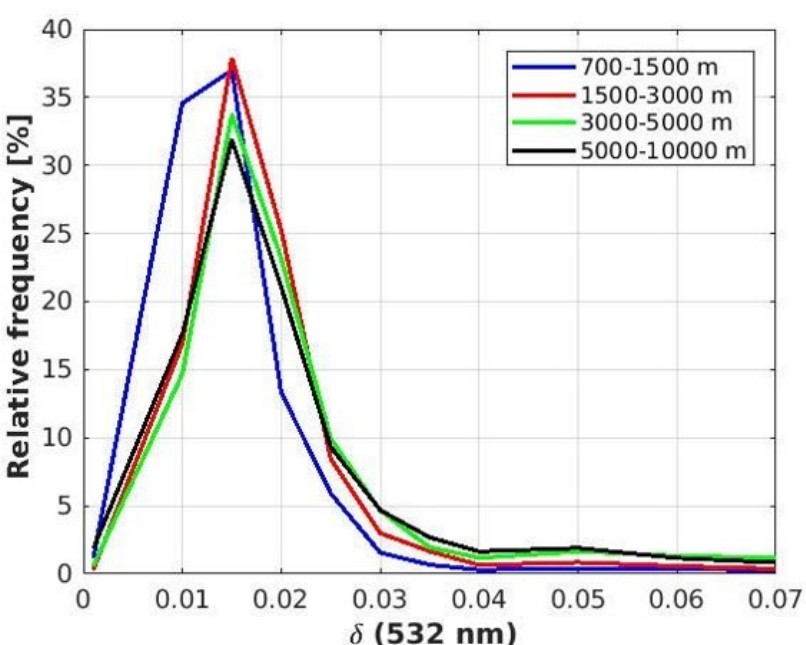

**Figure 6.** January–April: Frequency of $\delta_{532}$.

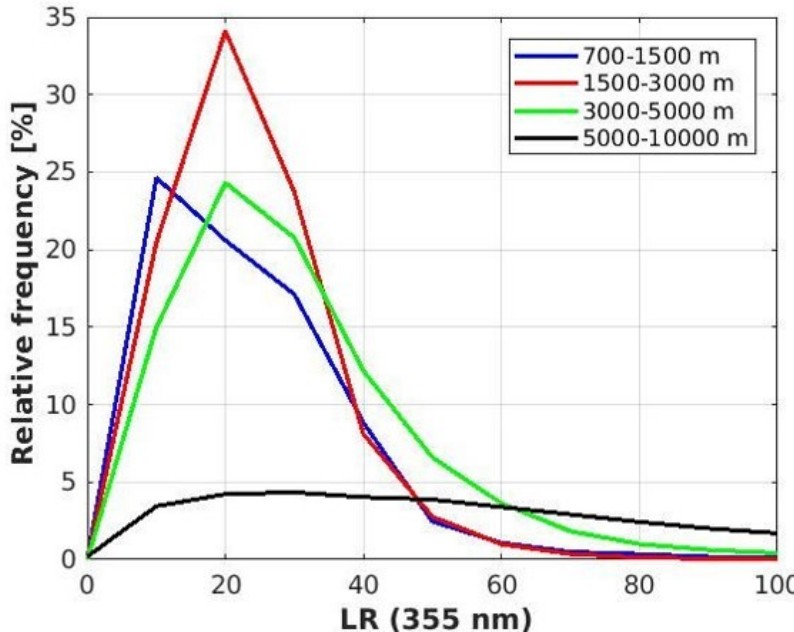

**Figure 7.** January–April: Frequency of the $LR_{355}$.

The colour ratio showed differences between the height levels. High colour ratio values were observed between 700 and 1500 m, indicating smaller particles. However, a broadening towards lower colour ratio values was observed for 3000–5000 m and 5000–10,000 m, indicating the presence of mixed particles with different sizes and, thus different ageing state or type. The CR can reach the Rayleigh limit of $\frac{\lambda_1}{\lambda_2}^{-4}$ when very tiny aerosol is present, which was observed in clean conditions above 3 km altitude. The graphs were cut off at this limit. The depolarisation was similar for all heights, with the interval 700–1500 m displaying slightly lower depolarisation values (more spherical particles). Therefore, the aerosol microphysical properties can be inverted using Mie theory as a good approximation and subsequently the aerosol radiative impact can be estimated. The lidar ratio at 355 nm was mostly low with values around 20 sr for the heights between 700 and 5000 m. Between 700 and 1500 m, the lowest lidar ratio values were observed, indicating non-absorbing aerosol

such as sea salt particles [40]. The LR between 5000 and 10,000 m was very broad, implying the presence of different aerosol types that were most likely aged and internally mixed.

### 3.2. Relation between the Optical Parameters and Relative Humidity (Rh)

In the following, we investigate aerosol hygroscopicity. This is done by examining the dependence of aerosol backscatter coefficient and colour ratio on relative humidity by contemporary radiosonde data. The height interval between 1500 and 2000 m was studied in order to see a correlation between different optical parameters. Below 1500 m, the aerosol is assumed to be variable due to surrounding mountains. By contrast, above 2000 m altitude the aerosol concentrations decreases so hygroscopic effects will be harder to see. Figure 8 depicts the relation between aerosol backscatter and colour ratio. It can be seen that generally the particles are small. (A colour ratio of 2.5 in our case translates into an Ångström exponent of backscatter of 1.81, which is larger than the photometer derived Ångstöm exponent [21] for Ny-Ålesund). Larger particles are nicely correlated to larger backscatter. This means that generally the size of the aerosol determines the backscatter (not, e.g., different chemical composition).

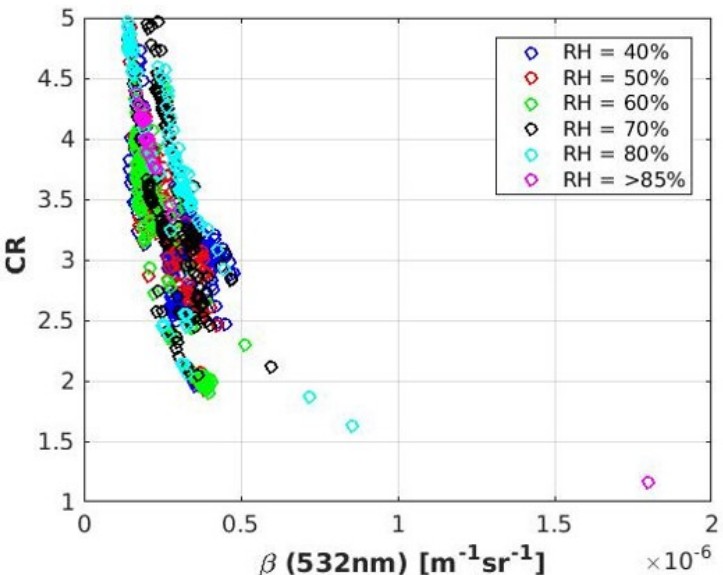

**Figure 8.** The colour ratio against the backscatter at 532 nm between 1500 and 2000 m altitude. The different colours indicate the corresponding RH ranges.

Subsequently, the relation between aerosol backscatter and RH from contemporary ($\pm30$ min) radiosondes was analysed. In total, 22 radiosonde and 1163 matching lidar profiles were used in this comparison. In Figure 9, it can be seen that high RH did not lead to larger backscatter. Especially the profiles with highest RH (above 85%) always showed backscatter coefficients below $0.4 \times 10^{-6}$ m$^{-1}sr^{-1}$. In total, 31% of the selected observations occurred at RH < 50%. Consequently, hygroscopic effects did not play a significant role for the late winter–early spring period of 2019. According to Figure 9, the most humid parts of the atmosphere were the cleanest in terms of aerosol load. Potentially some of the hydrophilic aerosol was already washed out prior to its advection to the Arctic or, alternatively, aerosol and moisture had different origins and pathways. The same can be concluded from Figure 10. Most of the aerosol found at RH > 85% had a colour ratio larger than 4, with this corresponding to very small particles. Only a few cases (~21.4%, in Table A2) and under very high RH conditions (RH > 85%) showed a colour ratio smaller than 2, indicative of hygroscopic growth. Detailed numbers belonging to Figures 9 and 10 are shown in Tables A1 and A2 in Appendix A.

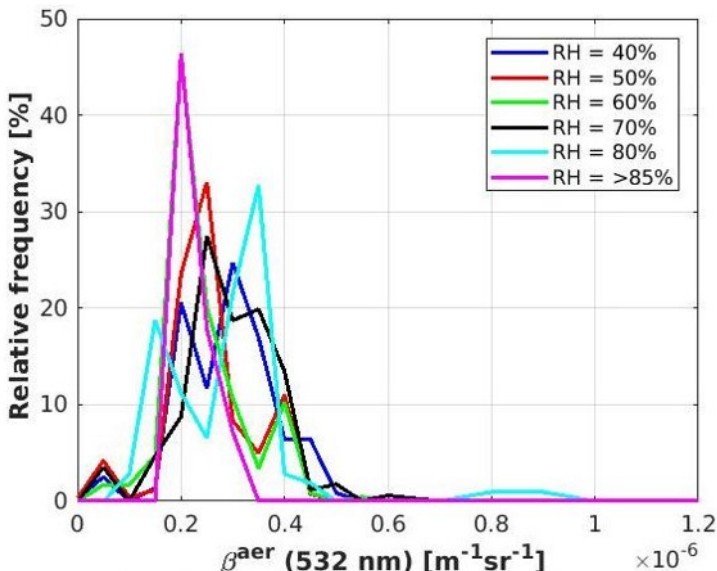

**Figure 9.** The backscatter at 532 nm for different values of the RH between 1500 and 2000 m altitude. The different colours indicate the corresponding RH ranges.

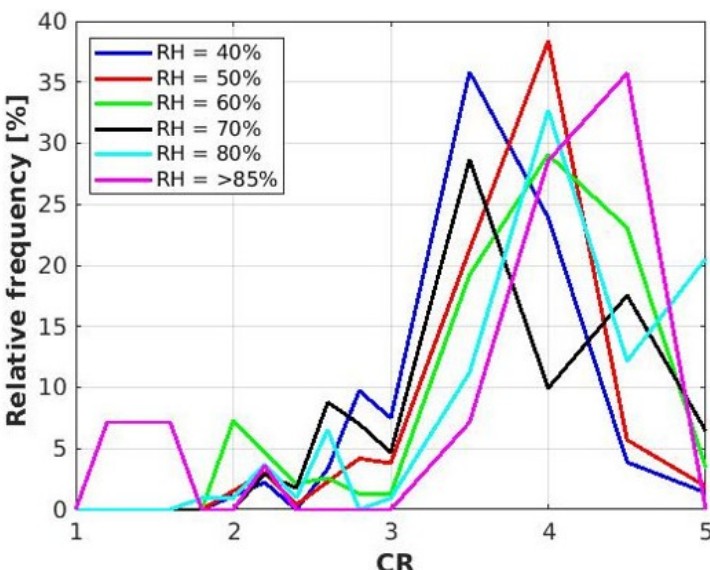

**Figure 10.** The colour ratio for different values of the RH between 1500 and 2000 m altitude. The different colours indicate the RH ranges.

### 3.3. In-Situ Measurements during Spring 2019

Figure 11 shows nss-$SO_4^{2-}$ concentration trends in aerosol samples collected at Gruvebadet and Zeppelin Observatories in the period January–June 2019 at daily–two day resolution. nss-$SO_4^{2-}$ is one of the main components of the *Arctic haze* and has been widely used as a proxy of this process in the high Arctic since many decades (e.g., Sharma [15] and Udisti [42]). Their concentration trends usually show a clear increase during the *Arctic haze* months peaking in March and/or April [12,16,38] and they have been used also to define the timing and characterise the Haze along the years. In 2019, surprisingly nss-$SO_4^{2-}$ does not show such a clear-cut signature, with a large occurrence of relatively high peaks from January to May. This pattern can be observed both at Gruvebadet and Zeppelin although the latter exhibits a lower variability ($0.509 \pm 0.500$ µg m$^{-3}$ and $0.477 \pm 0.413$ µg m$^{-3}$). In order to better appreciate the possible seasonal changes, distribution plots are reported in Figures 12 and 13. Despite a waving pattern in median values at Gruvebadet (ranging

between 0.310 and 0.695 µg m$^{-3}$), half of the values (box amplitude) spans a broad but comparable range between January and May (between 0.15 and 1.0 µg m$^{-3}$), to drop more clearly only in June (0.10–0.25 µg m$^{-3}$. At Zeppelin this sort of "steady-state" pattern during the Haze months is even clearer, with very close median values from January to May (0.20–0.48 µg m$^{-3}$) and especially very similar data distribution, which shows again lower values only in June.

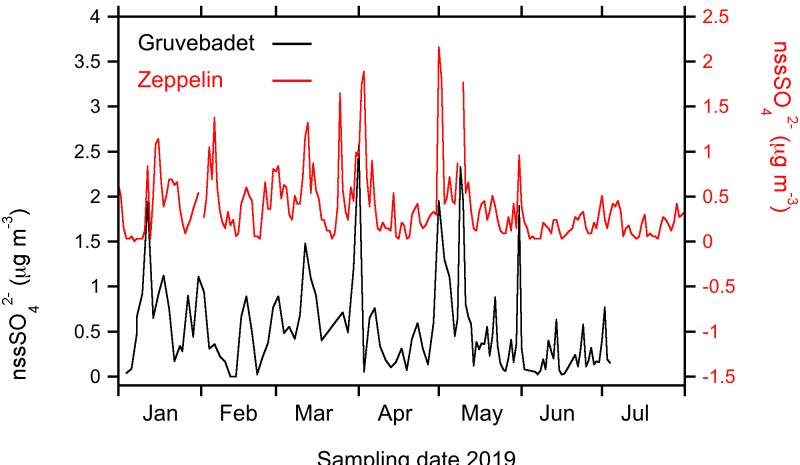

**Figure 11.** nssSO$_4^{2-}$ concentrations for spring 2019 at the village (Gruvebadet) and the mountain (Zeppelin) station.

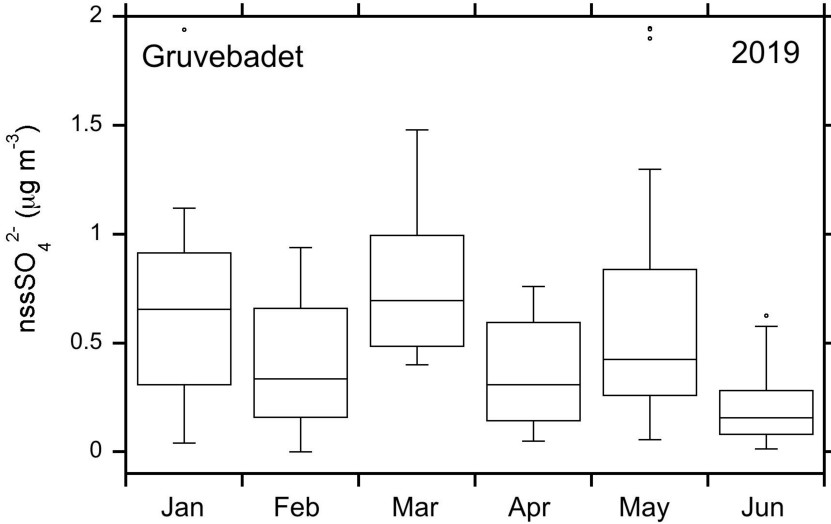

**Figure 12.** SO$_4^{2-}$ concentration from Gruvebadet station.

The peculiar features of 2019 *Arctic haze* can be observed also in Figure 14, showing the source apportionment of sulphate accomplished for Gruvebadet on the basis of ion ratios in sea salt, crustal and biogenic sources [16]. It can be seen that anthropogenic sulphate covers most of the budget of nss-SO$_4^{2-}$ (include also biogenic and crustal sulphate) in the whole period from January to May, except for a few days in mid May, where biogenic source accounts for at least half of sulphate concentration, as expected for this time of the year [44]. Starting from early June, anthropogenic sulphate drops while biogenic and sea salt inputs become more important. As anthropogenic sulphate can be related to the *Arctic haze* and/or other long-range transport processes, it appears that either the Haze has surprisingly lasted all through May or different processes (for instance, local meteorology) are taking place and controlling sulphate concentration so that no "classical *Arctic Haze*

period" can be actually detected in this year at Ny-Ålesund. In the latter case, we should probably revise the classical definition of *Arctic haze*, especially if such a pattern will be observed in the upcoming years.

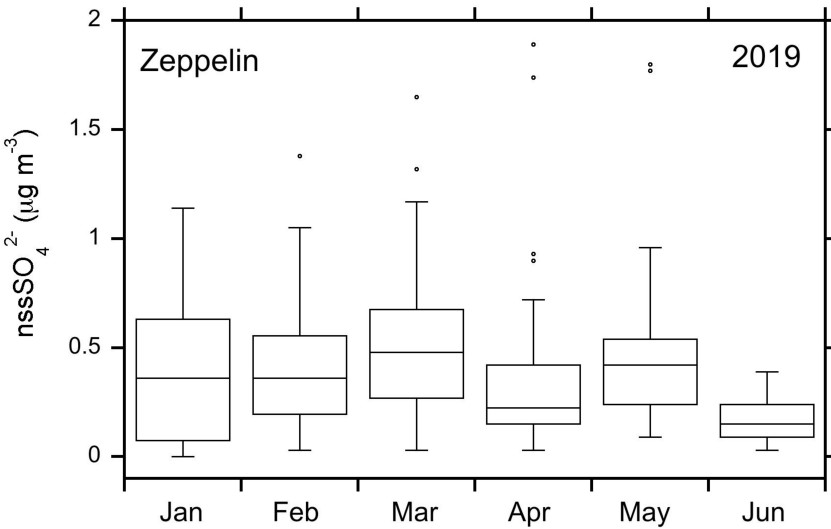

**Figure 13.** $SO_4^{2-}$ concentration from Zeppelin station.

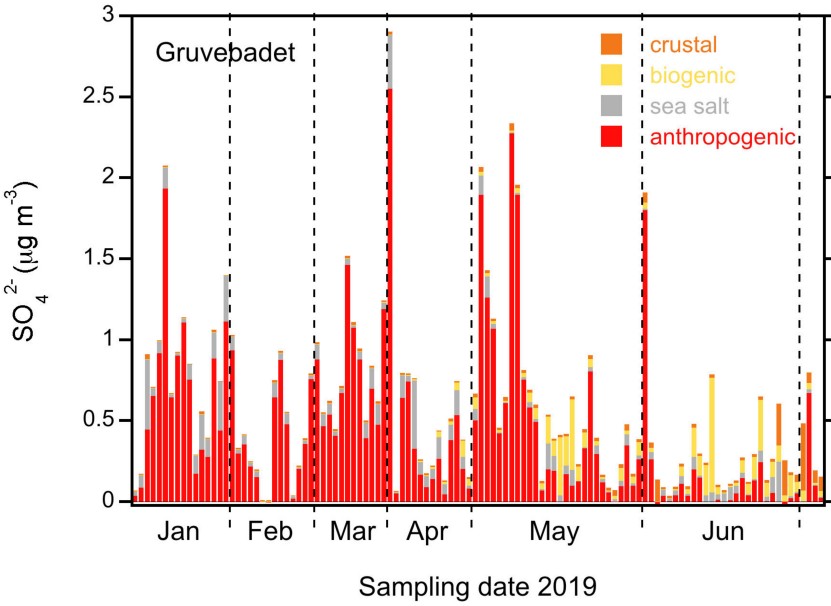

**Figure 14.** Source apportionment of $SO_4^{2-}$ from Gruvebadet Station.

## 4. Discussion

### 4.1. Comparing Lidar Data 2019 with 2018 and 2014

In order to assess the variability of *Arctic haze* on longer time scales, we compared the findings of this study (season of 2019) with the aerosol conditions during the years 2018 and 2014.

Müller [45] evaluated Lidar data from 1st of March to 13th of March 2018 over the site of Ny-Ålesund. That data were used to compare with all available March 2019 data (5, 7, 21–23, 25–28 and 30–31 of March). Figures 15–17 show the comparison for the backscatter, the colour ratio and depolarisation ratio (see Tables A3–A5 in the Appendix B).

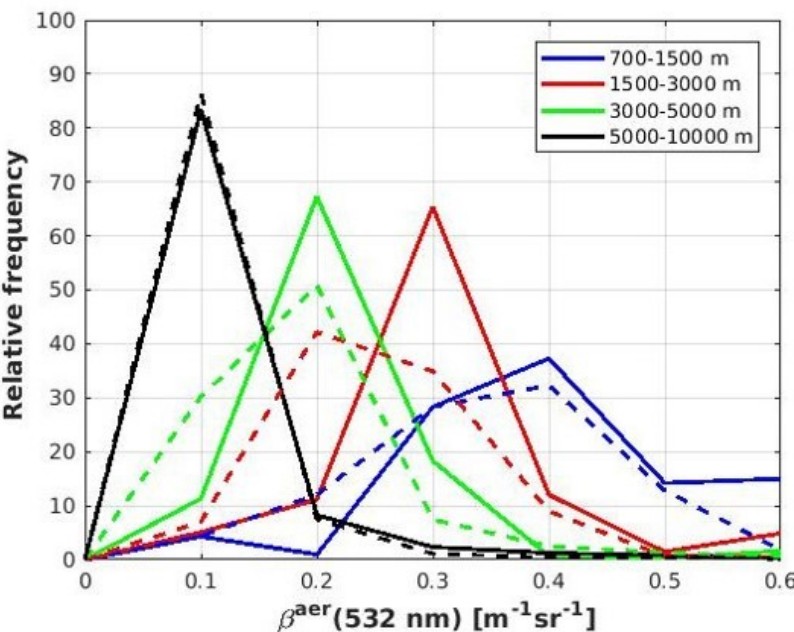

**Figure 15.** Aerosol backscatter data from 1 to 13 of March 2018 (Müller [45]) in comparison to the data from this study (all available data from March 2019). The filled lines (in *italic*) represent the 2018 data, while the dashed lines represent the 2019 data (this study).

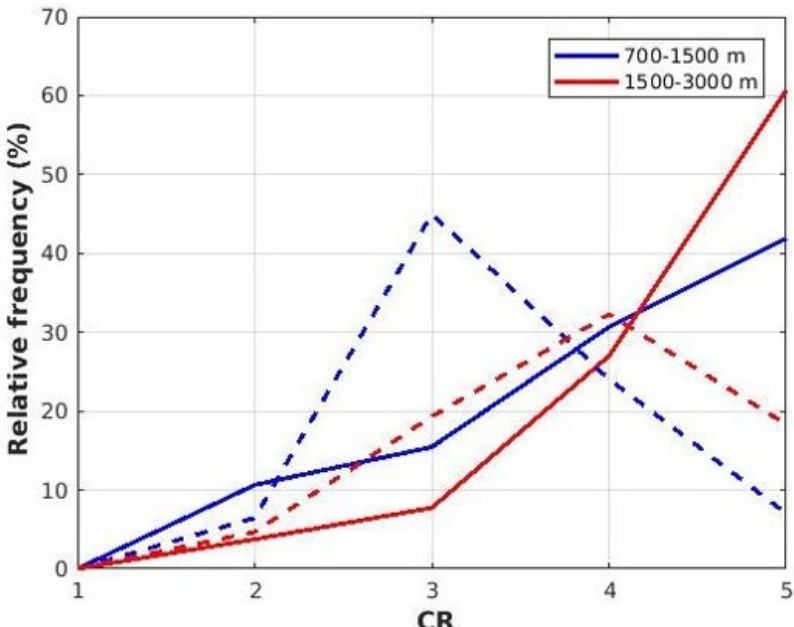

**Figure 16.** Comparing the colour ratios from 1 to 13 of March 2018 (Müller [45]) in comparison to the data from this study (all available data from March 2019). The filled lines (in *italic*) represent the 2018 data, while the dashed lines represent the 2019 data (this study).

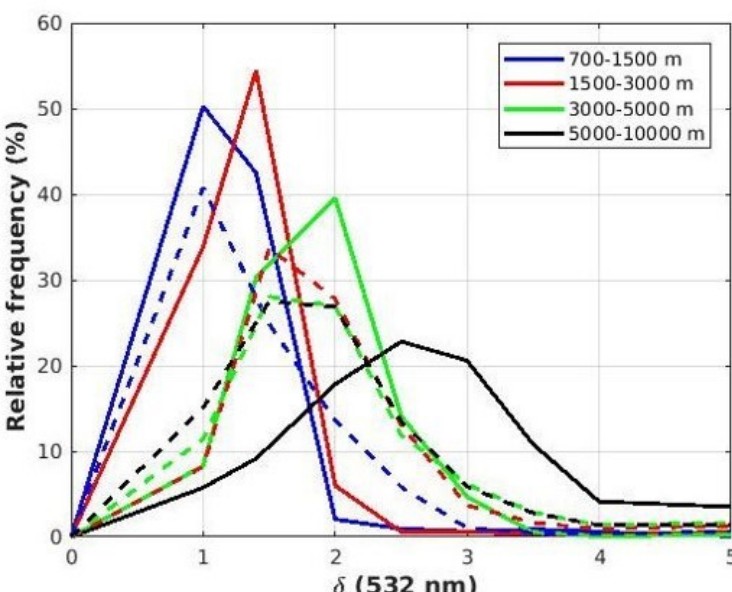

**Figure 17.** Comparing the depolarisation ratios from 1 to 13 of March 2018 (Müller [45]) in comparison to the data from this study (all available data from March 2019). The filled lines (in *italic*) represent the 2018 data, while the dashed lines represent the 2019 data (this study).

Between 700 and 5000 m altitude, higher aerosol backscatter coefficient values were observed in 2018. The largest differences were found for the height interval between 1500 m to 3000 m altitude. Above 5000 m the atmosphere was very clear in both years, as low backscatter with almost identical backscatter frequency distributions was observed. The depolarisation was generally low and indicates almost spherical particles. Slightly lower depolarisation values were found below 5000 m for 2018. Above that altitude the conditions changed and the depolarisation was larger in 2019. Therefore, the 2019 data suggest a depolarisation which rises with altitude while in 2018 this trend is far less obvious. The colour ratio between 700 and 1500 m seemed to be more uniform in 2019 and showed a broader distribution with a tail of small particles in 2018. Between 1500 and 3000 m the particles were larger (lower CR) during 2019, despite the lower backscatter.

The lidar data of 2019 were also briefly compared to those derived during the iAREA campaign in April 2014. During the iAREA campaign the aerosol backscatter coefficient was mostly around $0.4 \times 10^{-6}$ m$^{-1}$ sr$^{-1}$ (1000–1500 m). This is similar to April 2019, as a peak around $0.4 \times 10^{-6}$ m$^{-1}$ sr$^{-1}$ was observed as well (700–1500 m). The depolarisation ratio in April 2014 peaked between 0.02 and 0.03 (1000–1500 m). In April 2019, the depolarisation ratio was mostly 0.01–0.02 (700–1500 m). Therefore, the aerosol backscatter coefficient patterns of April 2019 were similar to those of the iAREA campaign. However, in this study the depolarisation ratio was slightly lower, indicating more spherical particles.

### 4.2. Comparing Different Sites around Ny-Ålesund

As an Arctic site, Ny-Ålesund typically faces a shallow and stable boundary layer in winter and early spring [31,46]. In reality, the boundary layer structure might be complicated due the orographically structured terrain of Ny-Ålesund that causes pronounced micrometeorology [31]. For this reason, we investigated the evolution of the boundary layer height on a daily basis. In Figure 18, the boundary layer height is shown for late winter and early spring 2019, as derived by the bulk Richardson number (using the local radiosounding data). The bulk Richardson number [47] considers profiles of potential temperature and wind and is a well suited criterion to determine the boundary layer height even under thermally stable conditions. The boundary layer during late winter–early spring 2019 was always shallower than 200 m, and especially low from January to March. In the majority of the study period (66%), the boundary layer top height was below 100 m,

with a median height of 70 m. The low Arctic boundary layer height clearly indicates that the lidar provides measurements (starting at 700 m altitude) above the local boundary layer and, thus, the reconciliation with ground-based in situ observations is not a straightforward task. There might even be different aerosol compositions between the Gruvebadet and Zeppelin site with less local, marine aerosol being present at the latter station.

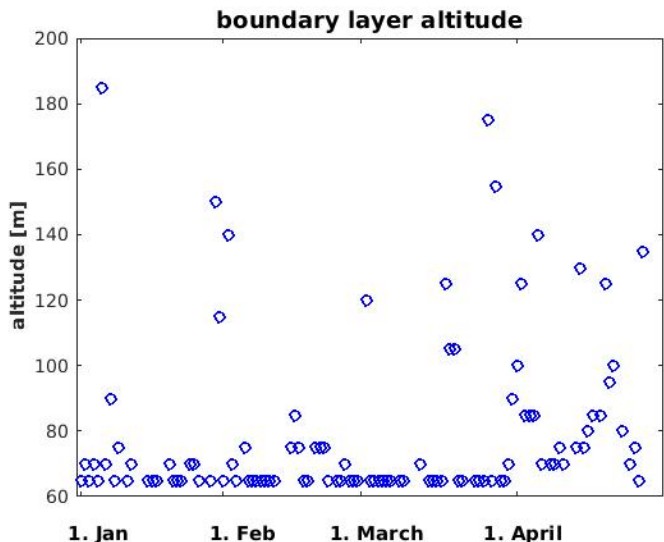

**Figure 18.** Boundary layer altitude from local radiosonde using the bulk Richardson number.

Figure 19 presents the correlation between the in situ measured nss-SO$_4^{2-}$ at the Zeppelin (mountain) and Gruvebadet stations. The same data as for Figure 11 are shown. Despite the low boundary layer height the correlation of nss-SO$_4^{2-}$ is almost 0.7 ($R^2 = 0.48$) and, considering the large data set (more than 100 values), it is highly significant (>99%). Furthermore, no systematic bias between the two sites can be seen (the linear fit slope is close to one). Therefore, neither of the stations systematically missed a fraction of the aerosol. However, as the nss-SO$_4^{2-}$ is expected to be long-range transported (there are hardly any local sources), the differences at the two in situ stations are remarkable. Our observations support the following mechanism: in late winter–early spring aerosol is advected into the Arctic in the lower free troposphere (higher wind speed and less vertical mixing compared to the ground). As reported by Thomas et al. [48], aerosol in the Arctic is mainly advected in lower altitudes during winter, while stronger boundary layer stability results in increased aerosol trapping below inversions. However, whether the aerosol will be finally collected by the ground-based in situ instruments depends on the downward vertical mixing, and therefore on the boundary layer stability. Therefore, the role of downward vertical mixing as a linking parameter between boundary layer and free tropospheric aerosol conditions are probably important in the Arctic.

While nss-SO$_4^{2-}$ is thought to be the most important component of Arctic haze [12,49], it will not be the only component on an Atlantic coastal site in years with below average pollution [16]. To estimate the importance of nss-SO$_4^{2-}$ during the winter–spring season 2019, the relation between PM10 and nss-SO$_4^{2-}$ is plotted in Figures 20 and 21. It can be seen that the correlation is not very high, especially for higher PM10 concentration the correlation gets poor.

The aerosol backscatter at 1–2.5 km exhibited less variability (correlation among the backscatter in the three layers of Figure 22 larger than 0.6) as qualitatively compared to the sulphate concentration (Figure 11). Even though both the ground-based and mountain Zeppelin stations revealed sharp sulphate concentration peaks between end of March and beginning of April (Figure 11), the lidar did not detect any aerosol backscatter peak despite its higher temporal and vertical resolution. Further, we note from the missing correlation

between lidar data and surface nss-SO$_4^{2-}$ that even in the altitude of the lidar measurements sulfate is probably still not the dominant aerosol component in our data.

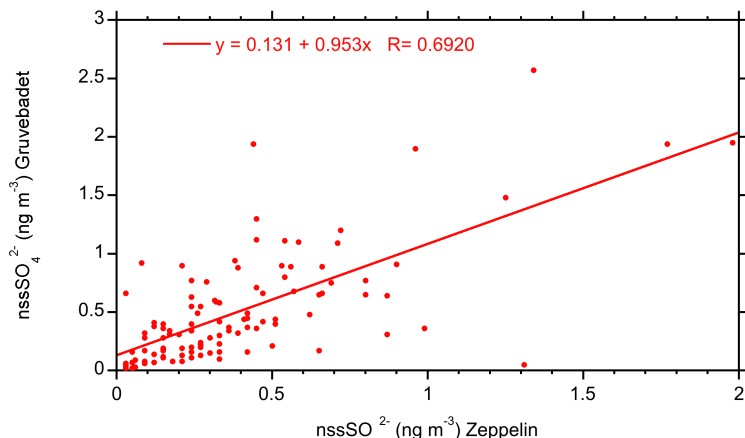

**Figure 19.** Correlation between nss-SO$_4^{2-}$ concentration at both in situ sites.

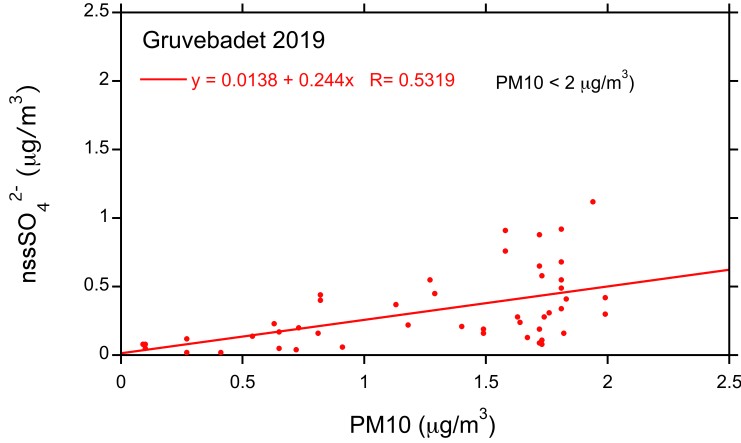

**Figure 20.** Relation between nss-SO$_4^{2-}$ concentration and PM10 from Gruvebadet station for low PM10 load.

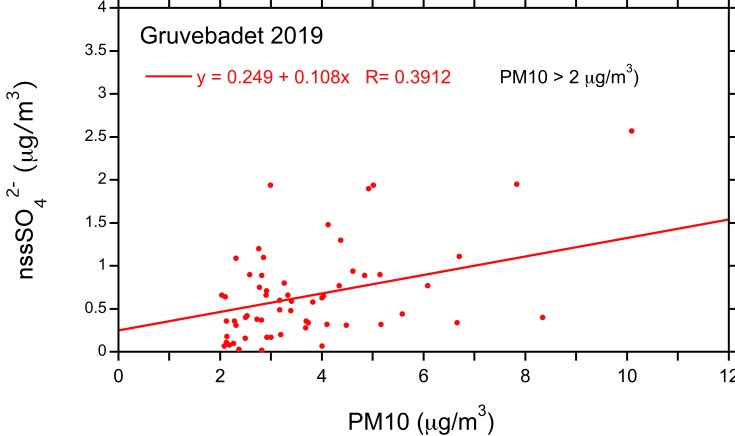

**Figure 21.** Relation between nss-SO$_4^{2-}$ concentration and PM10 from Gruvebadet station for high PM10 load.

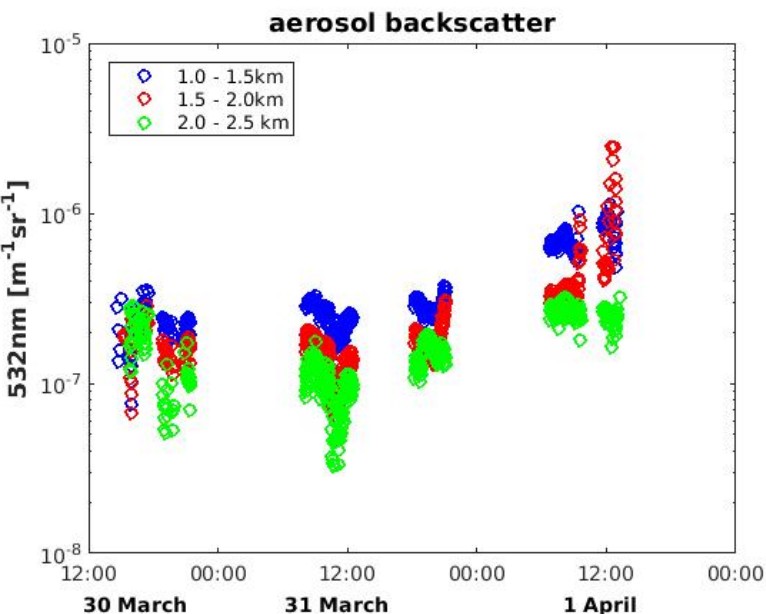

**Figure 22.** Averaged aerosol backscatter in three different altitude intervals.

In conclusion, although the backscatter coefficient between 700 and 1500 m varied only by a factor of 3 (Figure 4, Table A3) the concentration of the main in situ aerosol species fluctuated by a factor of 25 (Figure 11). Therefore, we hypothesise that the fraction of long-range transported aerosol measured on ground was heavily influenced by boundary layer dynamics such as downward vertical mixing. This effect is expected to be more pronounced in sites with heterogeneous orography as Ny-Ålesund. By contrast, in the free troposphere the aerosol properties mainly become a function of large scale advection and synoptic meteorological conditions. Considering that the transport of nss-$SO_4^{2-}$ from mid-latitudes into the Arctic takes about one week in winter [50], rapid changes in aerosol properties, in the order of hours, are only expected under highly variable meteorological conditions such as changing wind direction. For the aforementioned reasons, the aerosol properties should exhibit lower temporal variability in the free troposphere compared to the ground, as observed in this study. If this were true, boundary layer dynamics would play a key role for the analysis of ground-based aerosol properties at remote locations at least for the European Arctic. Likewise, Thomas et al. [48] highlighted that vertical aerosol distribution depends on the strength of temperature inversions.

## 5. Conclusions and Outlook

In this study, we analysed lidar-derived aerosol observations in combination with in situ non-sea salt sulfate measurements. Our investigation focused on the remarkably aerosol clear period of late winter–early spring 2019. As the measurements were obtained from three different sites—Ny-Ålesund (lidar), Gruvebadet and Zeppelin (in situ)—embedded in complex orography, we placed special emphasis on their reconciliation feasibility. The main findings of this study can be summarised as follows:

- The late winter–early spring season of 2019 was clear, with lower aerosol backscatter coefficient, especially in the altitude from 1.5 km to 3 km and lower non-sea salt sulphate concentration compared to previous years [13,14,27,51]. In contrast to other years, the aerosol backscatter in the free troposphere did not increase during March and April, the otherwise peak months for *Arctic Haze*. Therefore, for the European Arctic site of Ny-Ålesund and from the lidar perspective, 2019 presented itself "as a year without obvious *Arctic Haze*". In the future, our findings can be compared with satellite lidar or ground-based observations from the American and Russian parts of the Arctic. Such a comparison could be used to answer the question, whether the (remain-

ing) Arctic Haze phenomenon is mainly governed by the sources (decrease depending on source region) or sinks of aerosol (dependent on local meteorological conditions).

- In situ measurements from the two nearby stations, on mountain Zeppelin and at Gruvebadet (sea level), compared well for long-range advected sulphate on a seasonal basis (slope close to one). However, daily nss-sulphate concentration only showed a correlation in the order of 0.7. Moreover, we expect differences in aerosol composition between the two in situ sites, with less local marine aerosol at Zeppelin station. Therefore, a combined assessment of aerosol chemical composition at the Gruvebadet and Zeppelin sites is needed in the future.

- Over Ny-Ålesund, the aerosol load changed by less than a factor of 3.5 above 700 m. Surprisingly, the daily sampled nss-sulphate concentration erratically changed by a factor of 25 (from 0.1 to 2.5 ng m$^{-3}$) both at Gruvebadet (ground level) and Zeppelin station (474 m a.s.l.), with the latter mostly lying above the boundary layer during the study period. Overall, spherical particles were observed by the lidar. In the higher troposphere, the aerosol backscatter coefficient was confined to low values, indicating longer temporal scales and less mixing with new air masses.

- A possible systematic bias between lidar and in situ measurements might be due to hygroscopic growth, which might partly be lost by warming and drying of the air flow in the inlets in Arctic conditions. However, no noticeable hygroscopic growth was found from synchronous lidar and radiosonde measurements. Higher than average backscatter values generally occurred at moderate relative humidity. Neither the aerosol backscatter coefficient nor the colour ratio showed any positive correlation to relative humidity. We conclude that obviously aerosol and moisture have different origins (pathways) and that part of the aerosol may have been washed out during its advection towards the remote site of Ny-Ålesund.

- Based on the lidar-derived uniform aerosol properties in the free troposphere and the high day-to-day variability of in situ-derived nss-sulphate concentration, we conclude that aerosol is mostly advected in the lowest free troposphere and mixed downward erratically into the shallow Arctic winter–spring boundary layer. Therefore, we hypothesize that the Arctic ground-based aerosol properties generally show higher temporal variability compared to the free troposphere. This implies that the comparison between lidar and ground-based in situ observations might be more reasonable on longer time scales, i.e., monthly and seasonal basis. The same holds true for the two in situ sites around Ny-Ålesund. Further studies on the boundary layer along the slope of Zeppelin mountain are needed to understand the reported differences in aerosol concentrations.

The complex orography may introduce implications in the comparison of point observations with regional climate models. It should be noted that the lidar site (Ny-Ålesund) and the two in situ stations (Gruvebadet and Zeppelin) are located only a few kilometers apart and, thus, within one grid cell of current regional climate models. Through comparing non-sea-salt sulphate concentration with lidar backscatter observations, we concluded that ground-based in situ and remote sensing data of the free troposphere around a site with complex orography generally does not agree. Therefore, the derivation of a sound observational aerosol product over Ny-Ålesund for input or validation of climate models remains an open task. Finally, the high climate sensitivity of the Arctic is of concern. Over the Zeppelin site, a shifting of aerosol from a polar towards a marine regime is already observed [52]. In combination with the intensified advection of warm and moist north Atlantic air masses [4] over Ny-Ålesund and the unusual warming by the west Spitsbergen current [53], the question whether current aerosol conditions will become typical in a warmer future Arctic becomes highly relevant.

Further, in the future, we aim at increased aerosol data coverage by extending our analysis to aerosol scenes with overlying cirrus clouds [54,55].

**Author Contributions:** The manuscript has been written by F.R. and was reviewed by R.T. and C.R. Data analysis was done by F.R. and K.-J.M. for the lidar under supervision of C.R. In situ data was analysed by M.S., S.B. and R.T. Data Curation was done by K.N. and S.B. All authors have read and agreed to the published version of the manuscript.

**Funding:** The authors did not receive any external funding.

**Institutional Review Board Statement:** Not applicable.

**Informed Consent Statement:** Not applicable.

**Data Availability Statement:** Not applicable.

**Acknowledgments:** The research activity at Gruvebadet was made possible by Projects PRIN-20092C7KRC001 and RIS 3693 "Gruvebadet Atmospheric Laboratory Project (GRUVELAB)" and by the coordination of National Council of Research (CNR), which manages the Italian Arctic Station "Dirigibile Italia" through the Institute of Polar Sciences (ISP). Wilfried Ruhe and the station crew at AWIPEV launched the radiosoundings and assisted the lidar operations. Four anonymous reviewers improved the paper by careful proofread.

**Conflicts of Interest:** The authors declare no conflict of interest.

## Appendix A. Relation between Optical Parameters and Relative Humidity

**Table A1.** Relative frequencies the aerosol backscatter at 532 nm between 1500 and 2000 m for different RH values.

| Beta Classes | RH = 40% | RH = 50% | RH = 60% | RH = 70% | RH = 80% | RH > 85% |
|---|---|---|---|---|---|---|
| $1 \times 10^{-9}$ | 0.28 | 0.38 | 0.00 | 0.00 | 0.00 | 0.00 |
| $5 \times 10^{-8}$ | 2.50 | 4.18 | 1.71 | 3.51 | 0.00 | 0.00 |
| $1 \times 10^{-7}$ | 0.00 | 0.38 | 1.71 | 0.00 | 2.80 | 0.00 |
| $1.5 \times 10^{-7}$ | 1.39 | 1.14 | 4.70 | 4.68 | 18.69 | 0.00 |
| $2 \times 10^{-7}$ | 20.56 | 23.57 | 46.15 | 8.77 | 11.21 | 46.43 |
| $2.5 \times 10^{-7}$ | 11.67 | 33.08 | 20.09 | 27.49 | 6.54 | 17.86 |
| $3 \times 10^{-7}$ | 24.72 | 8.37 | 10.68 | 18.71 | 21.50 | 7.14 |
| $3.5 \times 10^{-7}$ | 16.94 | 4.94 | 3.42 | 19.88 | 32.71 | 0.00 |
| $4 \times 10^{-7}$ | 6.39 | 11.03 | 10.26 | 13.45 | 2.80 | 0.00 |
| $4.5 \times 10^{-7}$ | 6.39 | 0.76 | 0.85 | 1.17 | 1.87 | 0.00 |
| $5 \times 10^{-7}$ | 0.83 | 0.00 | 0.00 | 1.75 | 0.00 | 0.00 |
| $5.5 \times 10^{-7}$ | 0.00 | 0.00 | 0.43 | 0.00 | 0.00 | 0.00 |
| $6 \times 10^{-7}$ | 0.00 | 0.00 | 0.00 | 0.58 | 0.00 | 0.00 |
| $7 \times 10^{-7}$ | 0.00 | 0.00 | 0.00 | 0.00 | 0.00 | 0.00 |
| $8 \times 10^{-7}$ | 0.00 | 0.00 | 0.00 | 0.00 | 0.93 | 0.00 |
| $9 \times 10^{-7}$ | 0.00 | 0.00 | 0.00 | 0.00 | 0.93 | 0.00 |
| $1 \times 10^{-6}$ | 0.00 | 0.00 | 0.00 | 0.00 | 0.00 | 0.00 |
| $1 \times 10^{-6}$ | 0.00 | 0.00 | 0.00 | 0.00 | 0.00 | 0.00 |
| sum | 91.67 | 87.83 | 100.00 | 100.00 | 100.00 | 71.43 |
| No. of data points | 360 | 263 | 234 | 171 | 107 | 28 |

**Table A2.** Relative frequencies of the colour ratio between 1500 and 2000 m for different RH values.

| CR Classes | RH = 40% | RH = 50% | RH = 60% | RH = 70% | RH = 80% | RH > 85% |
|---|---|---|---|---|---|---|
| 1.0001 | 0.00 | 0.00 | 0.00 | 0.00 | 0.00 | 0.00 |
| 1.2 | 0.00 | 0.00 | 0.00 | 0.00 | 0.00 | 7.14 |
| 1.4 | 0.00 | 0.00 | 0.00 | 0.00 | 0.00 | 7.14 |
| 1.6 | 0.00 | 0.00 | 0.00 | 0.00 | 0.00 | 7.14 |
| 1.8 | 0.00 | 0.00 | 0.00 | 0.00 | 0.93 | 0.00 |
| 2 | 1.11 | 1.52 | 7.26 | 0.00 | 0.93 | 0.00 |
| 2.2 | 2.22 | 3.04 | 4.70 | 2.92 | 3.74 | 3.57 |
| 2.4 | 0.00 | 0.38 | 2.14 | 1.75 | 0.93 | 0.00 |
| 2.6 | 3.33 | 2.28 | 2.56 | 8.77 | 6.54 | 0.00 |
| 2.8 | 9.72 | 4.18 | 1.28 | 7.02 | 0.00 | 0.00 |
| 3 | 7.50 | 3.80 | 1.28 | 4.68 | 0.93 | 0.00 |
| 3.5 | 35.83 | 21.29 | 19.23 | 28.65 | 11.21 | 7.14 |
| 4 | 23.89 | 38.40 | 29.06 | 9.94 | 32.71 | 28.57 |
| 4.5 | 3.89 | 5.70 | 23.08 | 17.54 | 12.15 | 35.71 |
| 5 | 1.39 | 1.90 | 3.42 | 6.43 | 20.56 | 0.00 |
| sum | 87.50 | 80.61 | 90.60 | 81.29 | 70.09 | 96.43 |
| No. of data points | 360 | 263 | 234 | 171 | 107 | 28 |

## Appendix B. Comparing Data from 2018 with 2019

Tables A3–A5 show data of March 2018 and March 2019 for the backscatter, depolarisation and colour ratio. The 2018 values are given in *italic*.

**Table A3.** Aerosol backscatter data from 1 to 13 March 2018 (Müller [45]) in comparison to the data from this study (all available data from March 2019). The first value in each cell (in *italic*) represent the 2018 data, while the second value represents the 2019 (this study). Note the fact that the values of this study do not sum up as 100%: this is because the data were filtered for clouds and those values have been left out in the table and graphs, but have been included while calculating the relative frequencies.

| β (532 nm) $(10^{-6} \ m^{-1} \ sr^{-1})$ | 700–1500 m | 1500–3000 m | 3000–5000 m | 5000–10,000 m |
|---|---|---|---|---|
| 0–0.1 | *4.30* | *5.09* | *11.29* | *83.39* |
| | 4.38 | 7.05 | 30.16 | 86.43 |
| 0.1–0.2 | *0.96* | *11.11* | *67.28* | *8.27* |
| | 12.02 | 42.12 | 50.85 | 7.50 |
| 0.2–0.3 | *28.31* | *65.42* | *18.30* | *2.33* |
| | 28.36 | 34.93 | 7.44 | 1.09 |
| 0.3–0.4 | *37.27* | *12.04* | *0.70* | *1.23* |
| | 32.27 | 8.97 | 2.47 | 0.46 |
| 0.4–0.5 | *14.17* | *1.46* | *0.30* | *0.82* |
| | 12.82 | 0.62 | 1.32 | 0.39 |
| 0.5–0.6 | *14.98* | *4.88* | *1.58* | *3.96* |
| | 1.76 | 0.48 | 0.68 | 0.22 |

**Table A4.** Comparing the depolarisation ratios of Müller (2018) of the 1–13 March 2018 with the data from this study, considering all available data for March 2019. The first value in each cell (in *italic*) represent the data of Müller (2018), the second value represents the data found in this study. The ranges of the second and third differed slightly in the study of Müller compared to this study.

| Range | 700–1500 m | 1500–3000 m | 3000–5000 m | 5000–10,000 m |
|---|---|---|---|---|
| 0–1.0 | *50.25* | *33.83* | *8.30* | *5.76* |
|  | 40.80 | 8.19 | 11.45 | 15.07 |
| 1.0–1.4 | *42.48* | *54.55* | *30.28* | *9.12* |
| 1.0–1.5 | 24.81 | 33.73 | 28.03 | 27.49 |
| 1.4–2.0 | *2.05* | *6.00* | *39.58* | *17.83* |
| 1.5–2.0 | 13.62 | 27.75 | 26.95 | 26.84 |
| 2.0–2.5 | *0.96* | *0.58* | *14.08* | *22.79* |
|  | 5.83 | 13.03 | 11.91 | 13.47 |
| 2.5–3.0 | *0.63* | *0.58* | *4.67* | *20.57* |
|  | 1.05 | 3.68 | 6.13 | 5.87 |
| 3.0–3.5 | *0.91* | *0.33* | *0.54* | *10.78* |
|  | 0.47 | 1.65 | 2.86 | 2.78 |
| 3.5–4.0 | *0.58* | *0.24* | *0.07* | *4.13* |
|  | 0.31 | 0.92 | 1.33 | 1.38 |
| 4.0–5.0 | *0.10* | *0.61* | *0.26* | *3.51* |
|  | 0.66 | 1.31 | 1.68 | 1.37 |

**Table A5.** Comparing the CR values of Müller (2018) of the 1–13 March 2018 with the data from this study, considering all available data for March 2019. The first value in each cell (in *italic*) represents the data of Müller (2018), the second value represents the data found in this study. Note the fact that the values of this study do not sum up as 100%: this is because the data were filtered for clouds and those values have been left out, but have been included while calculating the relative frequencies.

| CR | 700–1500 m | 1500–3000 m |
|---|---|---|
| 1–2 | *10.64* | *3.75* |
|  | 6.46 | 4.67 |
| 2–3 | *15.44* | *7.75* |
|  | 44.98 | 19.31 |
| 3–4 | *30.66* | *26.98* |
|  | 24.06 | 32.26 |
| 4–5 | *41.92* | *60.65* |
|  | 7.04 | 18.32 |

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
