# Peer review of "Overview of Aerosol Properties in the European Arctic in Spring 2019 Based on In Situ Measurements and Lidar Data"

_atmosphere, doi:10.3390/atmos12020271_

Round 1
Reviewer 1 Report
Please find my comments attached.

Reviewer 2 Report
The abstract should be more concise, and I suggest authors provide the background, target, significance, methodology, main results, and so on, in this abstract.
The structure of the manuscript is inadequate. The manuscript should have sections: introduction, materials and methods, results, discussion and conclusion.
The introduction section is poorly organized, there are many missing links, and in addition, it is good to express the need for the study with the backlog of literature exists in the framework. There is no background in this introduction stating the urge and novelty of the study in which innovative ideas must be flown through the background along with the useful insights. What is the novelty for this article compared with existing studies? This manuscript does not present the purpose of the research.
The second section is not providing much insight information of the problem instead simply dealing with the basic things exist with the literature. Again, most of the literatures are not enough to hold the research gap of the study.
Literature review not well organized, I have never seen any research gap as a sub section, which is very important for proving the novelty of the paper. If possible, try to add research question in Introduction section, which may assist the readers to catch the point of the paper in the first section itself.
Discussion - This section is missing. The discussion should summarize the main finding(s) of the manuscript in the context of the broader scientific literature and address any limitations of the study or results that conflict with other published work.
The conclusion section should provide the main conclusions, not a summary and values of the results. Please revise. Also, it would be worth adding the limitations of the proposed approach and future research directions.
Reviewer 3 Report
This paper reads like a data report. The overall motivation is weak. What is the impact of your study? Throughout the manuscript, check the proper use of the definite article "the", i.e., it is not needed in some places, while it should be added in other places. There is a clear chance to improve English. Do not mix grammars (past and present). So many figures; you can combine some of them.
- Line 1: Add the definition of PM10 in the bracket. Do subscript “10”. Correct it throughout the manuscript.
- Line 2: “in-situ data were” Why “were”?
- Line 3: “Quite unusual”?? What were those? You can write it here.
- Line 5: “normal increase”??? How much increase? Write it in %.
- Line 6: “The two in-situ stations” Why “the”? Not necessary.
- Line 6: “recorded a a” correct it.
- Line 8: “about 60 %” There should not be a space in between the number and % sign. Correct it throughout the manuscript.
- Lines 8-9: “agreed very well” how much? Write the factor.
- You can write the abstract in the past tense. Check and improve English. Rewrite and make it more impressive.
- Line 31: 78.9N, 11.9E? The degree sign is missing.
- Line 60: There should be a space there. 387 nm, 607 nm
- Line 101: “other major ions” What are those? Write it here.
- Line 103: “is described in Becagli” changes to “is described elsewhere [26]”.
- Line 113” Delete “season”. Change the sentence structure. It’s confusing. You may write like this. “There was no presence of Arctic haze during the study period based on lidar data”.
- Line 119-121: “During 120 Jan to April 2019 the lidar derived aerosol properties are only a function of altitude, not of time.” Discuss more. Why so??? Change the sentence structure.
- The vertical distribution is analyzed in the next section.
- Line 166: non sea salt sulphate. You have already abbreviated this before. Do not repeat.
- Lines 182-184 and Figure 14: How did you do this source apportionment? Have discussed it in the method section?
- Line 200: “2018 and 2019 show similar trends” What is this????? No meaning.
- Line 200: “between 700 to 5000 m” Do you mean altitude. Please clearly mention.
- Line 200: “2018 shows” It was showed in 2018. Please correct the grammar.
- Line 201-202; 5000 m is was very clear..What is this??? Please correct the grammar.
- The conclusion is too big. Make it precise and compact. You can write it in bullet forms. Emphasize and include your highlights.
Reviewer 4 Report
This work is devoted to aerosol measurements by a sampler and a lidar in the Arctic, corresponds to the subject of the journal, and can be published after careful consideration of the reviewer comments presented below.
It is not clear from the text whether PM10 is a device or an aerosol parameter. If this is a device, it is called a PM10 sampler, not PM10 samples (line 1); it is unclear than what does it mean "PM10 aerosol" (line 86)?
Line 116: It is unclear to the reviewer how “the aerosol depolarization for the height interval 700m to 1500m” was calculated? Does it mean the aerosol depolarization averaged over the altitude range from 700 m to 1500 m? If so, it is necessary to indicate the number of measurements and the confidence intervals of the results shown in the figures.
Reference [22] (line 62) is unavailable to the reviewer, and the value of the laser power emitted at each wavelength is uncertain: if it is the average power, it is very high; if it is the peak power, it is too low.
Line 100: What is it 18 MOhm cm? I guess that this is the water parameter, resistivity, but it would be better to explain it.
Abbreviation “nss-sulphate” (line 9) is defined only in the Conclusion (line 265); please, define it when you first mention it.
Round 2
Reviewer 1 Report
Thank you for your answers and the modifications introduced to the manuscript.
Reviewer 2 Report
I have no other comments
Reviewer 3 Report
The authors have responded properly and adequately. The paper can be published now.